# Sustainability of Service Intermediary Platform Ecosystems: Analysis and Simulation of Japanese Hotel Booking Platform-Based Markets

**Yuki Inoue \*, Takeshi Takenaka and Koichi Kurumatani**

Human Augmentation Research Center, National Institute of Advanced Industrial Science and Technology, 6-2-3 Kashiwanoha, Kashiwa, Chiba 277-0882, Japan
\* Correspondence: yuki.inoue@aist.go.jp

**Abstract:** To achieve both a large number of platform users and the sustainability of service intermediation platform ecosystems, this study attempts to clarify appropriate platform fee settings and promising market types. It focuses on hotel-booking platforms and platform-based markets. We conduct agent-based simulation experiments with the conditions of (a) platform fee setting, (b) supply and demand balance, and (c) the consumer type categorized as "mandatorily" purchasing consumers (e.g., business travelers whose purchase of a hotel service is mandatory) and "optionally" purchasing consumers (who can refuse services if their requirements are not satisfied). Our simulation results reveal that when the platforms focus on mandatorily purchasing consumers, they could acquire larger platform users through fee settings, depending on the balance of supply and demand; however, they could not maintain ecosystem sustainability in any of the cases. When the platform focuses on optionally purchasing consumers, it can achieve both a large number of platform users and ecosystem sustainability with platform fee settings of consumer incentives, especially in excessive supply markets.

**Keywords:** platform-based market; platform ecosystem; two-sided market; service industry; hotel-booking platform; online travel agency; agent-based simulation

---

## 1. Introduction

Platform-based markets have a huge influence on human life in current society. Researchers have developed the concept of platform ecosystems [1,2]. Platform ecosystems are communities orchestrated by platform providers, outside complementors, and consumers [1–4]. A "business ecosystem" is an "economic community supported by a foundation of interacting organizations and individuals—the organisms of the business world; the member organisms also include suppliers, lead producers, competitors, and other stakeholders" [5,6]. A platform ecosystem restricts its scope to relevant actors, such as the platform itself as well as its providers and users. However, platform ecosystems do not generally restrict participation or withdrawal. Therefore, although a platform ecosystem has boundaries, it is an open system; that is, its composition may not mandatorily converge to any specific state because of outside interaction [7].

In platform ecosystems, complementors develop and/or provide complementary goods (products and/or services) using platform technology. Consumers then purchase these complementary goods via the platform. A platform ecosystem induces consumers with various needs to adopt the platform [8]. The success of a platform ecosystem depends upon the success of the entire ecosystem [9].

In fact, some previous studies have reported examples of platform ecosystems that declined earlier than those of competitors, although they succeeded in acquiring a large number of platform users.

For example, the Nintendo Wii video game platform ecosystem declined earlier than its competitor (PlayStation 3) due to the profitability of complementors (video game software providers) [10]. Furthermore, Nintendo DS declined faster than PlayStation Portable because of similar reasons [11]. Thus, in the research on platform ecosystems, we consider both their sustainability and growth as significant factors. Nevertheless, such research is insufficient.

Thus, this study examines the sustainably of platform ecosystems. In particular, we focus on service intermediary platforms. Typical examples of such platforms are Expedia and Hotel.com, which function as intermediates between hotel service providers and travelers as consumers. Such platforms can improve matching between consumers and services by allowing consumers to find preferable services with ease. If a platform can acquire a sufficient consumer base in the market, the service providers become interested in joining it, with an increasing expectation of obtaining numerous customers. The platform providers profit by charging fees for use of the platform during transactions.

As noted, one reason for the loss of sustainability in platform ecosystems is the profitability of its participants. General participants would not be motivated to remain with the platform at low profit. Therefore, one of the most significant factors for sustainability is profitability for complementors and consumers. At this point, we deem that service platforms may have two factors influencing sustainability, which may not occur in hardware–software-type platforms.

First, the earnings model of platform providers may be different. In case of hardware–software-type platforms, consumers have the incentive to purchase hardware because, generally, some parts of the software can be exclusive to the platform. Therefore, the platform provider can obtain profit from consumers. However, at least some current service intermediary platforms are different. For example, hotel-booking platforms typically handle existing hotel services. Consumers can use the same hotel services when not using the platforms. Therefore, we consider charging a platform fee to consumers to be difficult; the platform provider will gain profit from complementors (hotel service providers). We confirm that the fee settings of current hotel-booking platforms correspond to those in Section 2.1.2. Second, service providers have limitations in producing goods in comparison to product providers because of the inseparability of production and consumption [12]. Thus, charging a platform fee for service providers can have a larger negative impact on their profitability than it does on product providers. In fact, some hotel service providers experienced declining profitability when using hotel-booking platforms, as Section 2.1.1 shows. Thus, the current structure of service intermediary platforms may essentially become unsustainable ecosystems in terms of risk associated with profitability of service providers.

We now focus on the sustainability of service intermediary platform ecosystems. Considering that the platform fee setting is insufficient, we focus on market types, in which the service intermediary platforms can have large value. Thus, to achieve both a large number of platform users and sustainability of the service intermediation platform ecosystems, this study aims to clarify the appropriate platform fee settings and promising market types. We investigate hotel-booking platforms as representative of service intermediary platforms. This study adopts an agent-based simulation approach to its analysis.

## 2. Materials and Methods

The structure of this section is as follows. To understand current situation of hotel-booking platform ecosystems, we present the results of investigation of such platforms in Japanese hotel markets in Section 2.1. In Section 2.2, we explain the hypotheses of this study. In Section 2.3, we present an overview of our agent-based simulation system. We explain the detailed settings of agents in Section 2.4. Section 2.5 describes the simulation experimental settings.

### 2.1. Current Situation of Hotel Booking Platform-Based Markets in Japan

To understand the status of hotel-booking platforms for hotel service providers, we investigated the usage status of several service providers and their corresponding fee settings in the Japanese market.

### 2.1.1. Platform Usage Status of Service Providers

To confirm the usage status of service providers of the platforms in Japan, we conducted a questionnaire survey. We selected a total of 1000 Japanese hotel firms by random sampling, and mailed them the paper questionnaires. We set the response period from 16 January 2017 to 14 February 2017 and obtained 221 responses.

The results reveal that about 89.1% of hotel firms used at least one hotel-booking platform. Even if we consider that the respondents may have more interest in hotel-booking platforms than non-respondents, we believe that a large proportion of these hotel firms use such platforms. Specifically, Figure 1 shows the distribution of the use rate of each channel for capturing customers, including platforms by hotel firms. The results show that the mean value of the use rate via hotel-booking platform was about 33.8%; via travel agency, about 18.0%; via own website, about 39.6%; and via other channels, about 8.6%. Accordingly, the major channels for capturing customers of hotel firms in 2017 were hotel-booking platforms and own websites.

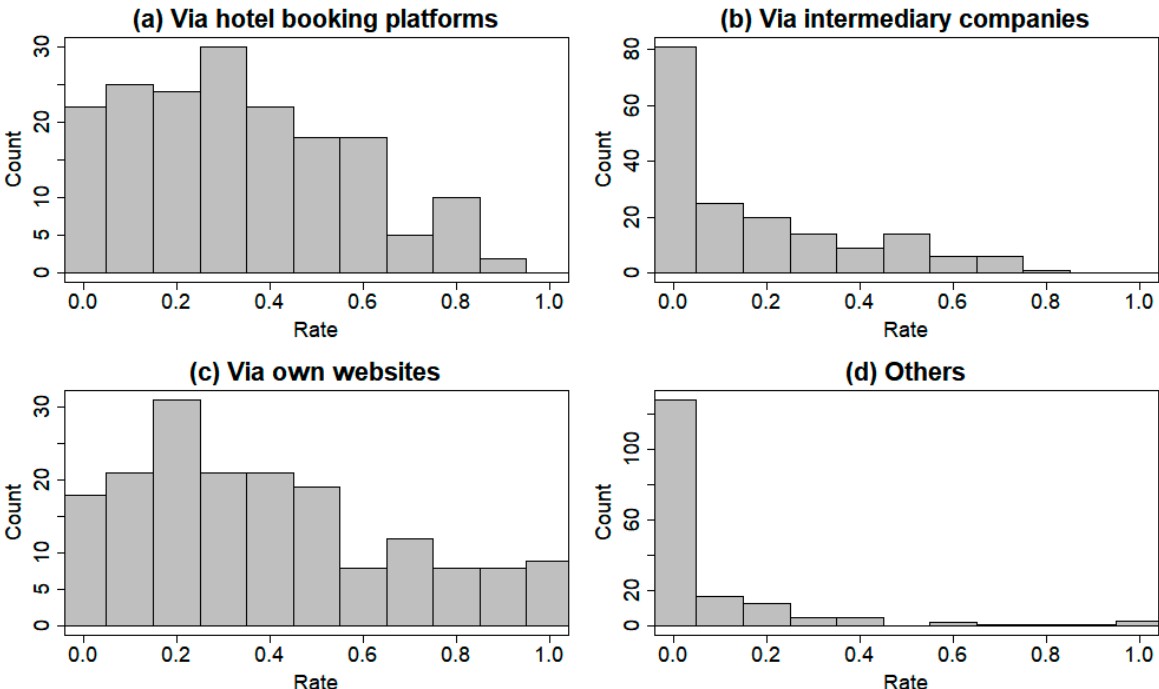

**Figure 1.** Use rate distribution of each channel for capturing customers by hotel firms, with 176 valid respondents to these items.

Figure 2 shows the evaluation results of the advantages and disadvantages of using hotel-booking platforms. About 81.7% of respondents agreed that "platforms can increase the number of new customers" (Figure 2a), and about 47.4% of them agreed that "platforms can increase the number of foreign customers" (Figure 2b). Meanwhile, about 63.1% of the respondents stated, "transaction fees of platforms are too high" (Figure 2c), and about 46.7% considered that "the platforms' transaction fees reduce their operating margin" (Figure 2d). In summary, we confirmed that although a large portion of hotel firms perceive benefits from the platforms, a significant portion also disagree with the platform fee and the perceived loss of profits because of such platforms in Japan.

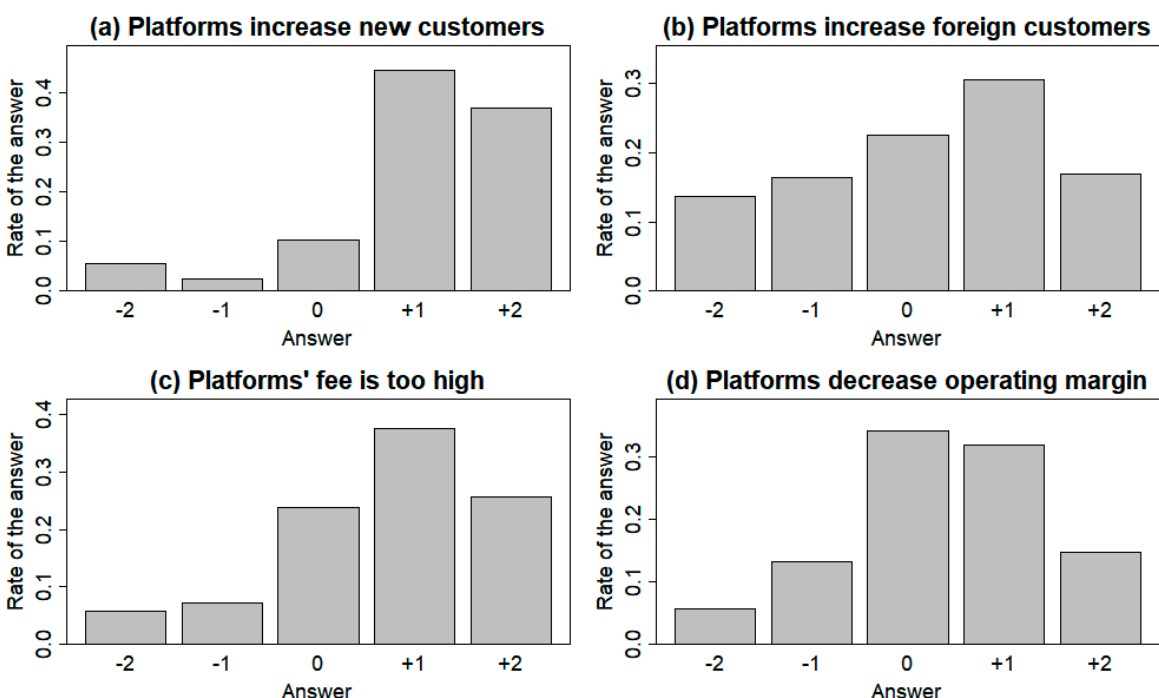

**Figure 2.** Evaluation results of hotel firms' perceived advantages and disadvantages using hotel-booking platforms. On the *x*-axis, the values correspond to the following: −2 = strongly disagree, −1 = disagree, 0 = neutral, 1 = agree, and 2 = strongly agree. The valid number of respondents is 213 for items (**a**,**b**) and 210 for items (**c**,**d**).

Additionally, Figure 3 shows the distribution of the room occupancy of hotel firms. The ratio values of most firms were within the 0.4–0.9 range, the mean value was about 0.61, and the mode value was 0.7. Accordingly, there was excessive supply in the Japanese hotel market, and spaces for market growth using the platforms remained.

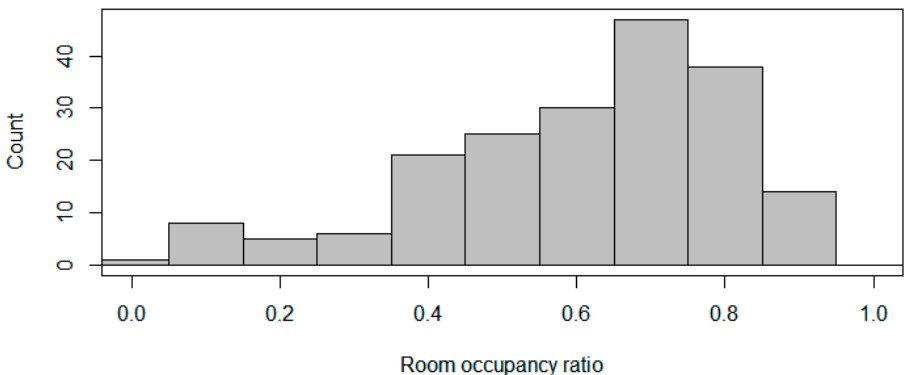

**Figure 3.** Distribution of room occupancy ratio of hotel firms with 195 valid respondents.

2.1.2. Fee Settings of Hotel-Booking Platforms in Japan

To confirm the fee settings of hotel-booking platforms in Japan, we collected information about platform fee settings from articles in Kanko Keizai Shimbun (newspaper on tourism and economy). These articles listed the primary hotel-booking platforms in Japanese markets from 2005 up to the access date (http://www.kankokeizai.com/site_survey/, accessed on 21 May 2019). Additionally, we calculated the portion of platform providers from the platform fee, as the difference between the higher and lower values within the platform fee for the hotel service providers and consumers.



We collected the latest information on fee settings for each platform. If the latest article contained no description of any platform fee settings, we collected information from past articles (from previous years to 2015). From the information, we confirmed that all platform fees were regarded as transaction fees, and not fixed fees. Moreover, we consider that the point system for consumers is a type of discount. Although some platforms allow hotel firms to set additional points at their own expense, we do not consider them in data collection. However, if some fee settings were adapted to the standard situation, we calculated their mean values. Furthermore, we removed non-standard hotel-booking platforms such as metasearch platforms (e.g., Trivago) and vacation rental intermediation platforms (e.g., Airbnb).

Figure 4 shows the distribution of platform fee settings and the portion of platform providers in the platform fee. In the platform fee settings, the negative values on the *x*-axis denote incentives. We obtained data from 23 platforms for the platform fee settings of service providers and 20 platforms for consumers. Our findings reveal that all platforms charged a fee to service providers and gave incentives to consumers. The mean value of service provider fees was 0.1 (10% of the transaction price), that of consumer incentives was −0.01 (1% return of the transaction price), and that of the platform provider share was 0.09 (revenue of 9% in each transaction price). Thus, we confirmed the fee settings of major hotel-booking platforms in the Japanese market as consumer incentive settings.

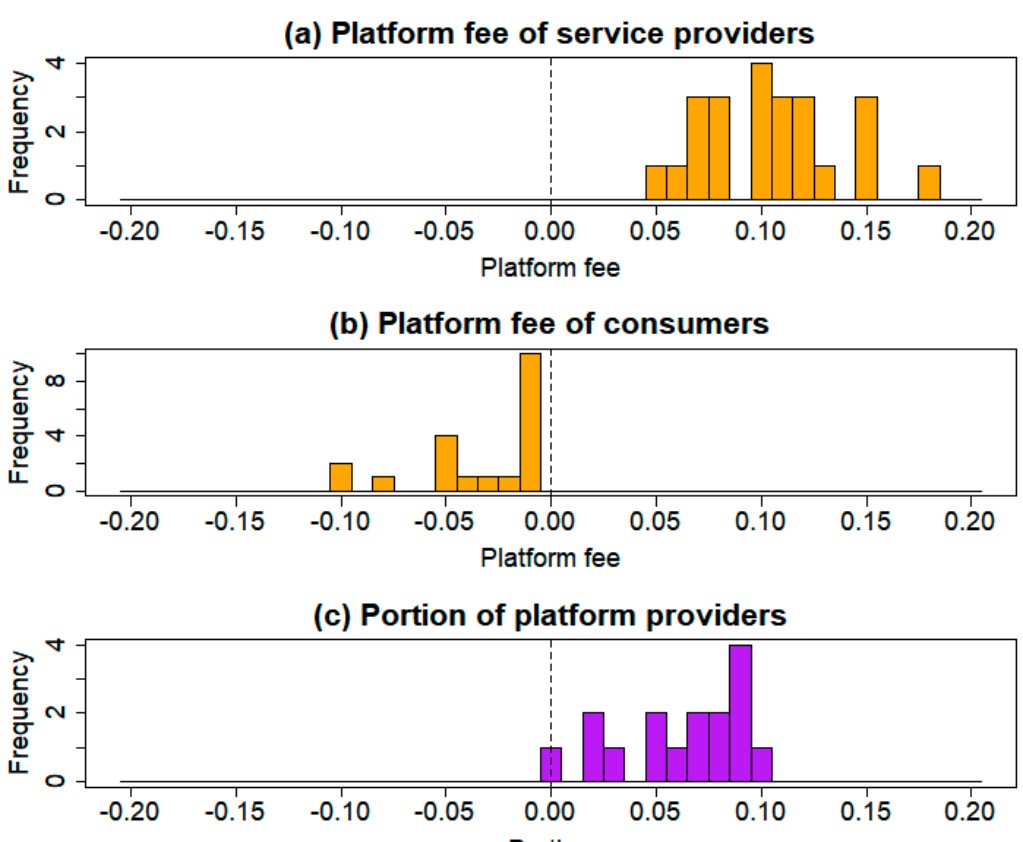

**Figure 4.** Distribution of platform fee settings and share of platform providers in the platform fee. In the platform fee settings, the negative values on the *x*-axis denote consumer incentives.

*2.2. Hypotheses Development*

The current platform fee settings converge on one dominant style. However, a non-negligible number of hotel service providers do not wish to pay this current platform fee and actually lose revenue. Accordingly, we aim to investigate more sustainable platform fee setting and market situations. This study considers the theories of previous studies and develops hypotheses for sustainable settings.

First, we develop a hypothesis from the pricing theory of two-sided platforms. For example, Yoo, Choudhary, and Mukhopadhyay [13] suggested that higher platform fees should be imposed on those who benefit from a more effective indirect network effect, indicating that the profit of participants on one side increases as the scale of the other side grows, in a two-sided market [14–18]. Following the Ramsey rule, Rochet and Tirole [15] stated that the platform price allocation should be based on its elasticity of demand. Caillaud and Jullien [19] suggested that the best strategy to dominate the market and protect market share is setting a low participation cost (possibly subsidized), along with maximum potential transaction fee. Meanwhile, Armstrong [20] discussed fixed and transaction charges and indicated that transaction charge allows participants to join a platform with lower expectations of transaction quantity than those with a fixed charge. Moreover, Armstrong [20] considered single homing and multi-homing and suggested that participants on the multi-homing side should be charged more than participants on the single-homing side. Rochet and Tirole [15] indicate a similar perspective on the pricing balance between single homing and multi-homing.

In summary, the following three factors would influence the appropriate platform fee settings:

(A)　Participants with higher demand (or greater indirect network effects) should be charged a higher platform fee.
(B)　Participants with a transaction charge would more easily join the platform than would those on a fixed charge.
(C)　Multi-homing participants should be charged a higher platform fee than single-homing ones.

If we consider the hotel booking platform-based markets, factors (B) and (C) would not be significant because most major current platforms adopt transaction fees, and both hotel service providers and consumers are multi-homing. However, we considered factor (A) because the demand balance between hotel service providers and consumers changes in seasonality, and this demand balance is captured as the balance between the size of service providers and that of consumers in hotel markets. Thus, we align these discussions with our objective and propose the following hypothesis.

**Hypothesis 1.** *Platform fee settings charged to participants in greater size can (a) facilitate more platform users and (b) achieve market sustainability.*

Second, we develop the hypothesis focusing on the function of hotel-booking platforms. The intermediary platforms, including hotel-booking platforms, appropriately arrange information and (explicitly or implicitly) recommend goods to consumers. As a recommendation method, we believe that consumer review [21] is the most typical for hotel-booking platforms. Consumers can select hotels that are more suitable by filtering the desired prices and evaluations of past consumers.

Such platform functions can also improve the purchasability of goods by reducing costs to consumers when searching for the goods. For example, search cost reduction might lessen consumers' confusion, thereby inducing the purchase. Information overload and very similar, complex, and ambiguous information can confuse consumers and exhibit the following behaviors [22]: (a) do nothing and ignore confusion, (b) abandon the purchase, (c) postpone the purchase, (d) clarify buying goals, (e) seek additional information, (f) narrow down the choice set by important criteria, (g) share the decision, and (h) delegate the decision. Apparently, behaviors (b) and (c) stop the purchase, whereas behaviors (d)–(h) generate consumer burden until the decision is made regarding purchasing of the goods. We consider that the platform functions may reduce the occurrence of behaviors (b) and (c) and may support behaviors (d)–(h) to solve consumer problems.

Here, we suppose that the effect of platform functions varies according to the type of consumers. Cho, Kang, and Cheon [23] suggested that highly quality-conscious consumers with uncertain needs, or who want to avoid wrong decisions, tend to become hesitant in completing online purchases. We consider that intermediary platforms could have more influence on these consumers. Conversely, intermediary platforms may not influence consumers with certain needs, or who make wrong decisions and can accept low-quality products because the existence of the platform did not change the purchasing

decision. In the latter context, we suggest the example of business travelers in the hotel booking platform-based market. Studies suggest that business travelers have the highest levels of expenditure, on average, among other types of travelers [24]. However, since business travelers must stay in any hotel for work (if the travel period is more than two to three days), the total quantity of hotel users should be constant regardless of the function of the platforms. By contrast, consumers without specific goals or tasks might also abandon their travel plans. If the platform functions succeed in supporting the plans of such consumers, it could increase market size, which is not limited to platform-based markets. In turn, the increase in profits for hotel service providers from market growth may be more apparent than the decrease in profit caused by platform fees.

We define consumers who must purchase goods as "mandatorily purchasing consumers" and consumers who can abandon purchases as "optionally purchasing consumers." We align these discussions to our objective and propose the following hypothesis.

**Hypothesis 2.** *If the platforms focus on the markets of optionally purchasing consumers, they can (a) facilitate a larger number of platform users and (b) achieve greater market sustainability than focusing on mandatorily purchasing consumers.*

*2.3. Overview of the Simulation*

Verifying the proposed hypothesis requires comprehensive data based on the combinations of platform fee settings and market situations. However, the current market cannot provide a sufficient empirical dataset. Accordingly, this study constructed an agent-based simulation system imitating transactions on hotel-booking platforms.

This agent-based simulation simulated the behaviors of actors (agents) who constitute the social system, especially in how they act to influence others [25]. The system is programmed on a computer, and the agent attempts to reproduce the real-life situations among the members by autonomously making decisions and interacting in the artificial environment [26]. The agent-based simulation approach is not a major technique in the field of strategy and management. However, some studies used this technique to determine the impact of new enterprises on the environment [27], the spread of innovation [28], the impact of consumer purchasing behavior on other consumers [29], the effectiveness of the management strategy of the retail chain store [30], the cooperative network formation in business ecosystems [31], and the competition between platforms [32]. The platform ecosystem in this study also consists of interactions between agents, and the outcome of the whole ecosystem can change owing to emergent phenomena caused by the interactions. Therefore, in this study, the application of the agent-based simulation approach is considered adequate.

In this study, following the simulation settings of Inoue, Takenaka, and Kurumatani [33], we constructed a simulation system. Moreover, to simplify the simulation settings, we added factors of change in market situations and evaluation indicators of market sustainability in the simulation. We built the simulation system with R language anew. This study set parameters and decision-making mechanisms of agents based on some assumptions and conditions supposing actual hotel-booking platforms. The simulation system includes three types of agents: platform provider, (hotel) service provider, and consumer agents. For simplicity of simulation, we set platform provider agent and the provided platform as single, although they exist as plural in the simulation environment. We also set each service provider agent to provide a single hotel service in the market. Each service provider agent (consumer agent) has inherent values related to service features (requirements for hotel service), including service price and service quality.

Both service provider and consumer agents can change their condition by either "using the platform and transacting with the opposite side (service provider agents or consumer agents) via the platform" or "not using the platform and transacting with the opposite side directly", as shown in Figure 5. The change of status is based on past profits in each method. Here, we assume that consumers

do not use any hotel-booking platforms to search and book hotels when transacting with the opposite side directly. In the real world, consumers can use internet search engines to access any hotel's website, and can book rooms directly. In our study, when consumer agents use the platform, they can refer to all the services from service pools on the platform. Conversely, when consumer agents do not use the platform, they can only refer to a limited number of services from the service pools not provided on the platform. Actually, almost all hotel-booking platforms have function sorting or extracting preferable services, which can support the consumers' searching ability and reduce time cost.

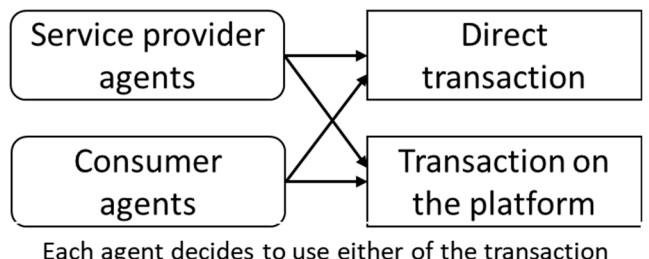

**Figure 5.** Decision-making by service provider agents and consumer agents.

The platform provider agent then charges a platform usage fee to service provider agents and/or consumer agents to sustain its business. The platform provider can give incentives to service provider agents or consumer agents, while it charges more fees to agents on the other side. We represent the incentive as a negative value of the platform usage fee in our simulation. For example, a platform fee of −0.1 means an incentive of 0.1. Instead of charging a platform fee, the platform provider intermediates in transactions between service provider agents and consumer agents. When service providers or consumers do not use the platform, they manage the transaction themselves. Figure 6 presents this relationship among agents in the simulation. The figure shows that the platform ecosystem includes platform provider agent, service provider agents using the platform, and consumer agents using the platform. Conversely, service provider agents not using the platform and consumer agents not using the platform are out of the ecosystem. Since the service provider agents and consumer agents can freely change positions, we express the platform ecosystem as an open system.

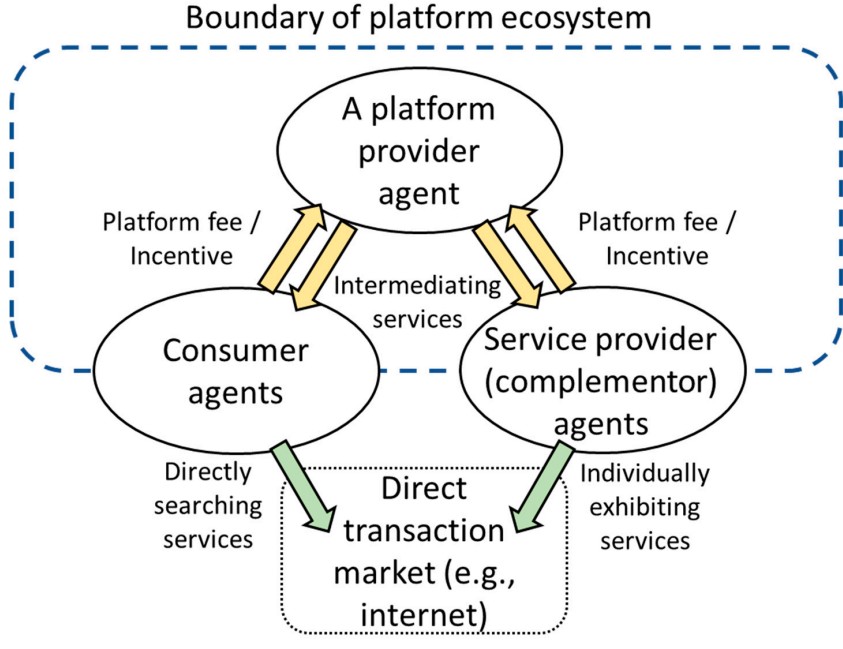

**Figure 6.** Relationship among agents in the simulation.

This study considers the time progress in the simulation. In each time step, each consumer agent searches and buys a service, and each service provider agent and each consumer agent decide whether to use the platform. Each agent takes over the situations of platform use in the previous one-time step and memory of acquired profits at previous time steps. Each time step includes the following procedure.

(a) Each consumer agent searches services and buys one service. Here, the consumer agent buys the service that provides highest profits among the selected services. The subsequent consumers cannot select and buy the already purchased service. When all consumer agents finish these processes, the profits of each agent in the step are calculated.

(b) Based on the acquired profits, service providers and consumer agents determine whether to use the platform for the next time step. To avoid obtaining results in any localized solution, this simulation uses the $\varepsilon$-greedy method at decision-making for platform use. In $\varepsilon$-greedy methods, agents select an optimal option at possibility of $1 - \varepsilon$, and randomly select one at possibility of $\varepsilon$. In this simulation, we set $\varepsilon$ as 0.1.

(c) For the next step, service provider agents set their services as available.

### 2.3.1. Conditions of Simulation Experiment

This study conducts simulation experiments by changing the conditions of platform fee and market situations as follows.

First, we set the platform fee as charged for each transaction, and not fixed with reference to actual hotel-booking platforms. The transaction fee is charged as the service price. The negative value of the rate denotes incentive. We set the range from −0.2 to 0.2, since the actual maximum platform transaction fee is about 20% of the service price, as in Figure 4. The simulation experiment can be conducted comprehensively by changing the platform fee of the service provider agents and consumer agents. As shown in Figure 4, we can confirm that the current portions of platform providers in platform fees are from 0 to about 0.1. Accordingly, we test the platform fee settings that satisfy the 0.05 portion for platform providers. Specifically, the combinations are as follows: (platform fee of service provider agents, platform fee of consumer agents) $\in$ {(0.2, −0.15), (0.15, −0.1), . . . , (−0.15, 0.2)}.

Second, related to Hypothesis 1, we test how the change in the balance of the number of service provider and consumer agents influences the simulation results. In our simulation experiments, we simply test two patterns of excessive supply and excessive demand. In excessive supply, the ratio of the number of service provider agents to consumer agents is 2:1 (in detail, 100:50). In excessive demand, it is set as 1:2 (in detail, 50:100). The normal, real world situation, as shown in Figure 3, is excessive supply. In peak seasons, it approaches excessive demand.

Third, related to Hypothesis 2, this study considers change in consumer purchases of hotel services. In detail, we test two patterns as mandatory purchase and optional purchase. As an example of mandatory purchase, when a consumer already has determined a travel or business trip, they must purchase a hotel service even if the price is high or the service quality is low. By contrast, as an example of optional purchase, when the trip is still being planned and can be withdrawn, the consumer will not purchase any hotel service if the price is considered expensive or the service quality is low. In our simulation, each service provider agent was provided with the service price and quality; similarly, each consumer agent was provided with the price requirement and quality requirement. In the mandatory purchase situation, consumer agents purchase a service even though it does not meet their requirement. By contrast, in the optional purchase situation, consumer agents purchase certain services that satisfy the price and quality requirement of the consumers.

Table 1 presents the summary of conditions of simulation experiment. Overall, we test 32 patterns of combinations.

### 2.3.2. Evaluation Indicators of Simulation Results

This study proposed two hypotheses as presented in the Introduction. To test these hypotheses, we set the indicators of "size of platform users" and "sustainability of the market in terms of profitability".

**Table 1.** Summary of conditions of simulation experiment.

| Conditions | Number of Patterns | Values and Specifications |
|---|---|---|
| Platform fee settings | 8 | (platform fee of service provider agents, platform fee of consumer agents) ∈ {(0.2, −0.15), (0.15, −0.1), . . . , (−0.15, 0.2)} |
| Balance of supply and demand of services | 2 | A: excessive supply (the ratio of the number of service provider agents to consumer agents is set as 100:50); B: excessive demand (the ratio of the number of service provider agents to consumer agents is set as 50:100) |
| Situation of consumers' purchase of services | 2 | α: mandatory purchase (consumer agents purchase any services even though the service does not satisfy their requirement); β: optional purchase (consumer agents purchase certain services that satisfy both the price and quality requirements of the consumers) |

First, we consider the indicator of platform user size, which has been calculated in platform-related research. One of the representative indicators is the size of the platform's installed base. For example, this value is calculated as the cumulative sales number of the platform. Establishing an installed base is important in generating indirect network effects [34–36]. A platform with a smaller installed base with no specialized markets would face negative growth through the indirect network effect [37]. However, the effectiveness of the installed base depends on the degree of influence of switching cost. Since the hotel-booking platform is mainly a web-based service and does not charge a membership fee, the switching cost may be too small to prevent the consumers' decision to use other platforms or other ways to search hotels. Accordingly, we should calculate the temporal platform user size, not the cumulative value of platform users. This study solved this issue by calculating transaction numbers between consumers and between service providers and consumers. Here, our simulation considered the change in situations of platform-based markets. We consider that such change in situations could influence the total transaction numbers. Therefore, this study calculates a transaction rate on the platform as an indicator of platform user size.

Second, we consider the indicators of ecosystem sustainability in terms of profitability. As stated in the Introduction, to the best of our knowledge, few studies have examined the sustainability of platform-based markets. For example, Inoue and Tsujimoto [10] indicated that the large new market of Nintendo Wii rapidly lost its sustainability because of the low profitability of software providers. Following this study, we focused on the increase and decrease in profits of participants on the platform-based market as an indicator of the sustainability of the market. In the case of hotel booking platform-based markets, the decrease in profits caused by the existence of the platform could lead to an increase in the discontinuance of hotel businesses. Similarly, a decrease in the profits of consumers due to the platform providers could reduce the interest of consumers in the travel markets. This study focuses on the sustainability of both service provider and consumer and defines the value of the market sustainability as follows. First, we calculate the differences between total profits on convergent results and total profits on non-platform results for service provider and consumer agent groups. Second, we define the indicator value of sustainability as follows: (a) it is set as 0 if the difference values for both groups is zero or less than zero; (b) it is set as 1 if the difference values for one groups is zero or less than zero, but another group is not; and (c) it is set as 2 if the difference values for both groups is more than zero. On the basis of these indicator values, this study regards an indicator value of 2 as indicating achievement of market sustainability.

### 2.4. Parameter Settings and Decision-Making of Agents

We explain parameter settings and specifications of decision-making for each type of agent as follows. Table 2 summarizes the variables in this study's simulation system.

#### 2.4.1. Platform Provider Agent

The profits gained by platform provider agent are dependent on platform fee, service price, and service sales on the platform. In our simulation, the platform provider agent does not make any decision and only intermediates services to consumers. The profit of the platform provider agent $v_t^P$ at the simulation step $t$ is calculated as

$$v_t^P = (r_{serv} + r_{cons}) \sum_k^{N_{comp}} p_k s_{k,1,t},$$

(1)

where $r_{serv}$ is the platform fee of service provider agents, $r_{cons}$ is the platform fee of consumer agents, $N_{comp}$ is the number of service provider agent, $k$ is the service provider agents ID ($k \in \{1, \ldots, N_{comp}\}$), $p_k$ is the service price provided by agent $k$, and $s_{k,1,t}$ is the sales quantity of service provided by agent $k$.

The platform provider agent provides the hotel-booking platform in the virtual market. The platform support consumers to search for services that are more suitable. We present this function in our simulation as follows. (a) When consumers do not use the platform, they can search for services from the service pool in the direct transaction market, due to ability and time cost limitations. We set our simulation such that consumer agents who do not use the platform can acquire three service selective candidates in each simulation step. Here, the candidates do not include services that are already purchased by other consumer agents, except when most services are already consumed. (b) By contrast, when consumers use the platform, they can search all services from the service pool in the platform-based market. In our simulation, we set that the consumer agents who use the platform can acquire the most preferable service that gives highest profit for the agent, excluding the already purchased services. This imitates the hotel-booking platform's functions of service sorting or extraction based on service price and quality (consumer review score).

#### 2.4.2. Service Provider Agents

Service provider agent $k$ can acquire profit when consumer agents purchase its service. Our simulation generates plural service provider agents; 100 agents are generated in case of excessive supply, and 50 agents in case of excessive demand. For simplicity, we set each service provider agent to provide one service to the market. Service provider agents change their status of platform use to improve their profits. In the initial step of the simulation, all of them are set as unused platform (direct transaction).

Each service provider agent has inherent values of service price $p_k$ and service quality $q_k$. Service price $p_k$ is set as random value based on standard distribution under mean of 10,000 and standard deviation of 1,250; that is, the value falls within the range of 5000–15,000. We set the values referring to Japanese hotel prices (Japanese Yen). Since service price and service quality are usually correlated, service quality $q_k$ is set as sum of $p_k/3000$ and random value, which is based on the standard distribution under mean of 0 and standard deviation of 0.25. We set the values referring to evaluation score of consumer review in actual hotel-booking platforms. The value is modified as satisfying the range from 0 to 5 and the sales profit of the service provider, exceeding zero.

**Table 2.** Summary of variables.

| Variable | Symbol | Values in the Simulation | Note |
|---|---|---|---|
| Simulation step | $t$ | $1, \ldots, 250$ | Since we confirm that our simulation results can converge until about 200 steps, we set the last step at 250 |
| Platform fee of service provider agents | $r_{serv}$ | −0.2, −0.15, −0.1, −0.05, 0, 0.05, 0.1, 0.15, 0.2 | Negative value of $r_{serv}$ denotes setting incentives for service providers |
| Platform fee of consumer agents | $r_{cons}$ | −0.2, −0.15, −0.1, −0.05, 0, 0.05, 0.1, 0.15, 0.2 | Negative value of $r_{cons}$ denotes setting incentives for service providers |
| ID of service provider agents (and their service) | $k$ | Situation of excessive supply: $1, \ldots, 100$ Situation of excessive demand: $1, \ldots, 50$ | Each service provider agent provides one service to the market |
| ID of consumer agents | $i$ | Situation of excessive supply: $1, \ldots, 50$ Situation of excessive demand: $1, \ldots, 100$ | Each consumer agent purchases one service at one simulation step |
| Service price | $p_k$ | Random value based on standard distribution with mean of 10,000 and standard deviation of 1250 | We set the values using Japanese hotel prices (Japanese Yen) |
| Service quality | $q_k$ | Sum of $p_k/3000$ and random value, which is based on standard distribution with mean of 0 and standard deviation of 0.25 | We set the values referring to the evaluation score of consumer review in actual hotel-booking platforms. The value is modified to satisfy the 0–5 range and the sales profit of the service provider exceeding zero |
| Maintenance cost of service providers | $f_k$ | $q_k \times 3000 \times 0.4$ | We set the value of 0.4 referring to the distribution of room occupancy ratio in the Japanese hotel industry |
| Requirement for service price | $p_i^D$ | Value multiplied by 1.1 and random value based on standard distribution with mean of 10,000 and standard deviation of 1250 | We set the mean value of $p_i^D$ to be slightly higher than that of $p_k$ |

**Table 2.** *Cont.*

| Variable | Symbol | Values in the Simulation | Note |
|---|---|---|---|
| Requirement for service quality | $q_i^D$ | Sum of $p_i^D/3000/1.1$ and random value, which is based on standard distribution with mean of 0 and standard deviation of 0.25 | We set the same distribution between $q_k$ and $q_i^D$ |
| Coefficients of standardization for price value in calculating consumers' profit | $\mu^P$ | 1 | Since this study unifies values of profit for all agents as unit of price, $\mu^P$ is set as 1 |
| Coefficients of standardization for quality value in calculating consumers' profit | $\mu^Q$ | 1/3000 | Since this study unifies values of profit of all agents as unit of price, the quality value is standardized as unit of price |
| Quantity of service sales | $s_{k,0,t}$ (on direct transaction), $s_{k,1,t}$ (on the platform) | 0 or 1 | The sum of $s_{k,0,t}$ and $s_{k,1,t}$ becomes 1 |
| Profit of the platform provider agent | $v_t^P$ | The values are calculated in the simulation process | Referring to Equation (1) |
| Profit of the service provider agents | $v_{k,t}^S$ | The values are calculated in the simulation process | Referring to Equation (2) |
| Profit of the consumer agents | $v_{i,t}^C$ | The values are calculated in the simulation process | Referring to Equation (3) |

Note: More details on each variable are covered in the following sections.

The decision-making of service provider agents for platform use/disuse is based on their profits at previous simulation steps under the circumstances of platform use and disuse. Profit $v_{k,t}^S$ of service provider $k$ at simulation step $t$ is calculated as

$$v_{k,t}^S = p_k\{s_{k,0,t} + (1 - r_{serv})s_{k,1,t}\} - f_k, \tag{2}$$

where $s_{k,0,t}$ is service sales quantity on the direct transaction, $s_{k,1,t}$ is service sales quantity of the platform, and $f_k$ is the running cost for service provision. Here, as shown in Figure 3, we confirmed that values of room occupancy ratio for most samples (88%) were above 0.4. Therefore, using this value as the basis for revenue balance and supposing that the running cost is dependent on the hotel and service quality, we define the running cost as $f_k = q_k \times 3000 \times 0.4$.

Each service provider agent has the information on expected profit $\hat{v}_{k,1,t}^S$ on the platform and expected profit $\hat{v}_{k,0,t}^S$ on the direct transaction based on past profits at previous simulation steps. In this simulation, we calculated the average value of the profits using the past four periods of each situation (platform use and direct transaction) as $\hat{v}_{k,1,t}^S$ and $\hat{v}_{k,0,t}^S$. Here, since all service provider agents start from the state of direct transaction, the values of $\hat{v}_{k,1,t}^S$ cannot be calculated. Considering the real world, we suppose that the service providers refer to the degree of participation of future customers and rival firms in deciding to use the platform. One of the influences of customers' participation on the platform is "indirect network effects." Conversely, one of the influences of rival firms' participation on the platform is "bandwagon effects," which is the organization's imitation of the other's adoption decisions regardless of efficiency or returns [38–40]. Indeed, a previous study indicated that the indirect network effects and bandwagon effects could induce participation of third-party firms to the platforms [11]. Referring to such previous studies, we set the initial value of $\hat{v}_{k,1,t}^S$ to be calculated as expected sales value $s_{k,1,t}$ in Equation (2), and regarded the mean value of use rate of consumer agents and service provider agents.

### 2.4.3. Consumer Agents

Consumer agent $i$ can acquire profits when purchasing the service from service provider agents. Our simulation generates plural consumer agents: 50 agents in the case of excessive supply and 100 agents in the case of excessive demand. For simplicity, we set each consumer agent to purchase one service at one simulation step. Consumer agents change their platform use status to improve their profits similarly to service provider agents. At the initial step of the simulation, all are set as platform disuse (direct transaction).

Each consumer agent has inherent values of requirement for price $p_i^D$ and requirement for service quality $q_i^D$. Supposing the mean value of service price becomes higher than the requirement price, we set values of $p_i^D$ as slightly higher than service prices $p_k$; the value is set as multiplied by 1.1 and random value based on standard distribution (mean is 10,000 and standard deviation is 1250). Similarly, for service quality, since the requirement of service price and service quality are usually correlated, we define that distribution of quality requirement $q_i^D$ corresponding to the distribution of service quality $q_k$; that is, $q_i^D$ is set as sum of $p_i^D/3000/1.1$ and random value, which is based on standard distribution under mean of 0 and standard deviation of 0.25.

The decision-making of consumer agents for platform use/disuse is based on their profits at previous simulation steps under circumstances of platform use and disuse similarity. In typical economic modeling, the user's net utility is determined by the difference between his/her willingness-to-pay and actual market price. However, previous researchers also considered that other factors could influence utility. For example, Berry [41] modeled consumers' utility function including not only price but also product characteristics and demand parameters. Indeed, if we think logically, consumers using hotel-booking platforms may consider some information (e.g., photos and reviews) about the quality of service when purchasing hotel services. Therefore, at least quality parameters should be included in the consumers' profit function. This study formulates this simply as the influence of satisfaction of

price requirement and of quality requirement are equivalent. Profit $v_{i,t}^C$ of consumer $i$ at simulation step $t$ is obtained as

$$v_{i,t}^C = \frac{p_i^D - (1 + r_{cons})p_k}{\mu^P} + \frac{q_k - q_i^D}{\mu^Q}, \tag{3}$$

where $\mu^P$ (or $\mu^Q$) is the coefficients of standardization for price value (or quality value) in calculating consumers' profit. Since the unit of profit values of other types of agents is price, we set the unit of consumers' profit as price and set the values as $\mu^P = 1$ and $\mu^Q = 1/3000$. If a consumer agent could not purchase any services, the value of $v_{i,t}^C$ is set as $-\infty$ (however, such a consumer is not included in the calculation of total profits).

Similarly, for service provider agents, each consumer agent has information on expected profit $\hat{v}_{i,1,t}^C$ on the platform and expected profit $\hat{v}_{i,0,t}^C$ on the direct transaction based on past profits in the previous simulation steps. In this simulation, we calculated the average value of profits of the past four periods of each situation (platform use and direct transaction) as $\hat{v}_{i,1,t}^C$ and $\hat{v}_{i,0,t}^C$. Since all consumer agents begin in the state of direct transaction, values of $\hat{v}_{i,1,t}^C$ cannot be calculated. Although we regard the size of participation of service provider agents and consumer agents on the platform as service providers agents' initial expected profit on the platform, such a size is difficult to translate into consumer profit, as in Equation (3). Considering that consumers decide to use new ways of searching for services, we suppose that their decision to use the platform stems from dissatisfaction with available services. Therefore, this study set the initial expected profit $\hat{v}_{i,1,t}$ to 0 on the platform as the neutral expectation value.

*2.5. Simulation Experiment*

There are 32 combinations of the conditions of simulation experiment, as summarized in Table 1. Since we confirmed the simulation could sufficiently converge at $t = 200$, we calculate the mean values from $t = 201$ to $t = 250$ as the simulation results. We test each combination 40 times and calculate the average values and standard deviations. We then present value of transaction rate on the platform and indicator value of sustainability as simulation results, as described in the Evaluation Indicators of Simulation Results section.

We summarize the simulation results corresponding to Hypotheses 1 and 2 in the Results section. To test Hypothesis 1, we show the results of correspondence between platform fee settings and the balance of supply and demand. In this test, we separately show the results of mandatorily purchasing consumer markets and optionally purchasing consumer markets. To test Hypothesis 2, we show the comparison of results between these markets. Furthermore, we separately show the results of situations of excessive supply and excessive demand.

## 3. Results

*3.1. Correspondence between Platform Fee Settings and Balance of Supply and Demand*

Figure 7 shows the simulation results of the test of correspondence between platform fee settings and the balance of supply and demand. The results of the platform transaction rate are in the upper row, whereas those of sustainability are in the lower row. Meanwhile, the results of mandatorily and optionally purchasing consumer markets are in the left and right columns, respectively. In each figure, the *x*-axis represents the platform fee settings. As the setting moves toward the left (right) direction, the incentives for consumers (service providers) become stronger.

First, we verify the results of the market with mandatory purchase. As Figure 7a-1 shows, in excessive supply, the platform transaction rate became high at the consumer incentive settings. Conversely, in excessive demand, the platform transaction rate became high at the service provider incentive settings. Therefore, in a market of mandatory purchase, Hypothesis 1a was supported. However, as Figure 7b-1 shows, the degree of sustainability was not higher than 1 in any case of mandatorily purchasing market; that is, either/both the profits of consumers or service providers were

decreased by the introduction of the platform in all cases. Therefore, for a market with mandatory purchase, Hypothesis 1b was not supported.

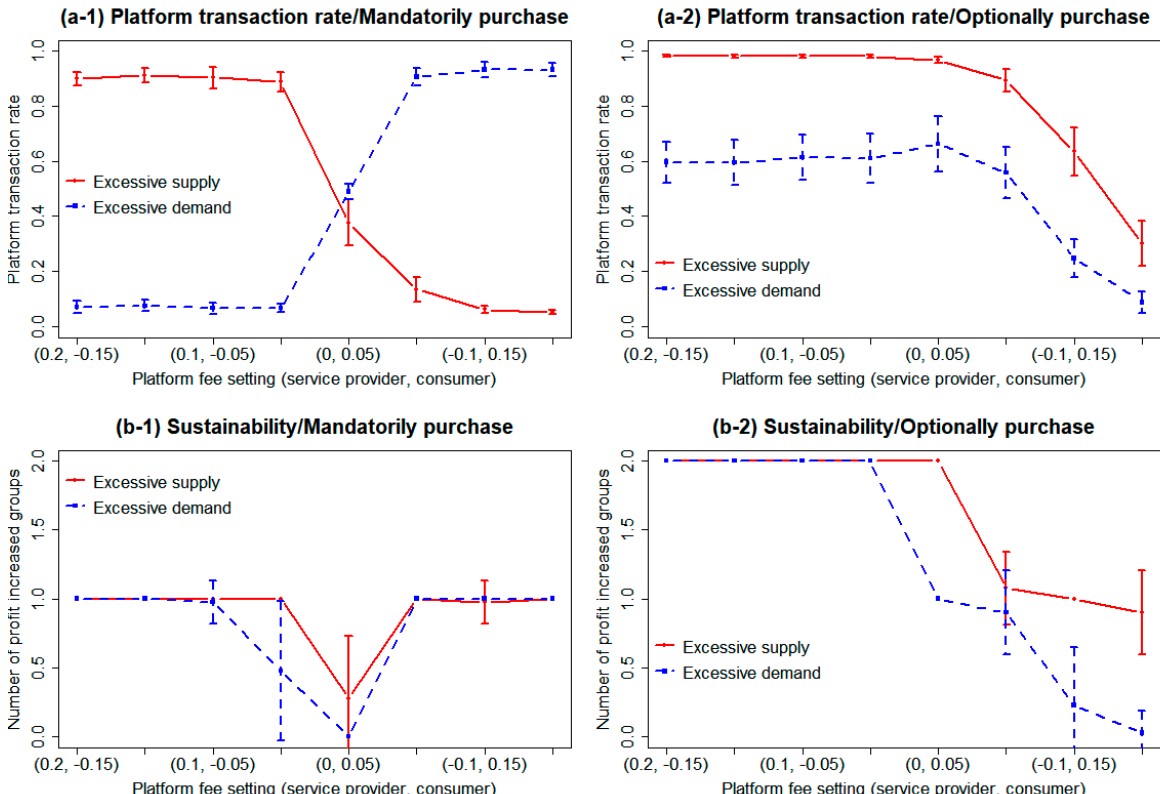

**Figure 7.** Simulation results on the test of correspondence between platform fee settings and balance of supply and demand.

Second, we verify the results of the optionally purchasing market. As Figure 7a-2 shows, in excessive supply, the platform transaction rate became high at consumer incentive settings. However, in excessive demand, the platform transaction rate did not increase at service provider incentive settings. Therefore, for a market with optional purchase, Hypothesis 1a was not supported. Our findings reveal that for a market with optional purchase, overall, consumer incentive settings had more advantage than the service provider incentive settings. Similarly, as Figure 7b-1 shows, the degree of sustainability had high value at the consumer incentive settings. Therefore, for an optionally purchasing market, Hypothesis 1b was not supported. Table 3 shows the summary of tests for Hypothesis 1.

**Table 3.** Summary of tests for Hypothesis 1.

|  | Markets of Mandatorily Purchasing Consumers | Markets of Optionally Purchasing Consumers |
|---|---|---|
| (a) Acquisition of a larger number of platform users | Supported | Not supported |
| (b) Achievement of sustainability | Not supported | Not supported |

*3.2. Comparison between Results on the Market of Mandatorily Purchasing Consumers and Those of Optionally Purchasing Consumers*

Figure 8 shows the simulation results of the test of comparison between the market of mandatorily purchasing consumers and those of optionally purchasing consumers. The results of the platform transaction rate and sustainability are in the upper row and lower row, respectively. The results in situations of excessive demand are in the right column. In each figure, we summarized the results of

consumer incentive settings ($r_{cons} < 0$) on the left side and results of service provider incentive settings ($r_{serv} < 0$) on the right side.

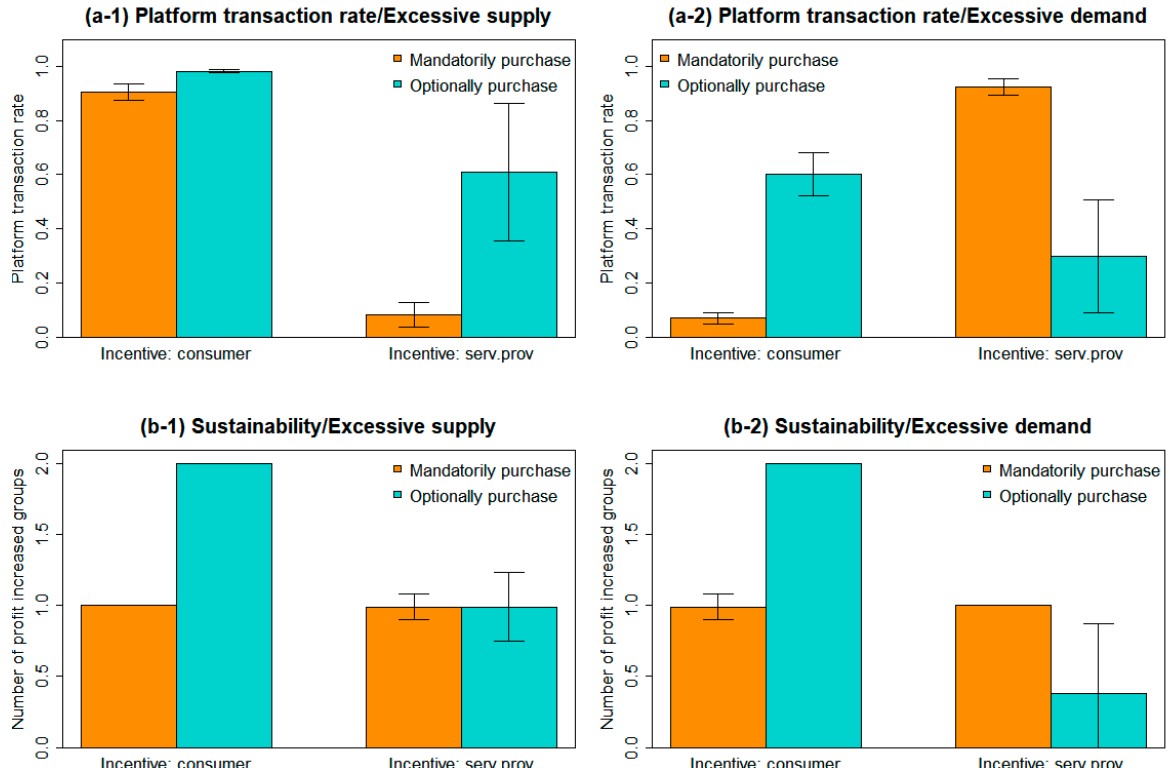

**Figure 8.** Simulation results of the test of comparison between results on the market of mandatorily purchasing consumers and those of optionally purchasing consumers.

First, we confirm the results of excessive supply. As shown in Figure 8a-1, in excessive supply, the platform transaction rate of optionally purchasing market tended to be higher than that of mandatorily purchasing market regardless of the platform fee settings. Therefore, in excessive supply, Hypothesis 2a was supported. As Figure 8b-1 shows, with consumer incentive settings, the degree of sustainability in the optionally purchasing market tended to be higher than in the mandatorily purchasing market. However, in excessive supply with service provider incentive settings, the indicator values became similar, on average, regardless of the purchase situations. Therefore, for excessive supply, Hypothesis 2b was supported at the consumer incentive settings but not at the service provider incentive settings.

Second, we confirm the results of excessive demand. As Figure 8a-2 shows, in excessive demand with consumer incentive settings, the platform transaction rate at optionally purchasing market tended to be higher than that at mandatorily purchasing market. However, for service provider incentive settings, the value at optionally purchasing market tended to be lower than that at the mandatorily purchasing market. Therefore, in excessive demand, Hypothesis 2a was supported at consumer incentive settings, but not at service provider incentive settings. Similar to these results, for excessive demand with consumer incentive settings, the degree of sustainability at optionally purchasing market tended to be higher than that at mandatorily purchasing market (Figure 8b-2). In the service provider incentive settings situation, this tendency was converse (however, the degree of sustainability of mandatory purchase is insufficient). Therefore, in case of excessive demand, Hypothesis 2b was supported at consumer incentive settings, but not at service provider incentive settings. Table 4 shows the summary of tests for Hypothesis 2.

**Table 4.** Summary of tests for Hypothesis 2.

| | Excessive Supply | | Excessive Demand | |
|---|---|---|---|---|
| | **Consumer Incentive Settings** | **Service Provider Incentive Settings** | **Consumer Incentive Settings** | **Service Provider Incentive Settings** |
| (a) Acquisition of a larger number of platform users | Supported | Supported | Supported | Not supported |
| (b) Achievement of sustainability | Supported | Not supported | Supported | Not supported |

## 4. Discussion

This study aimed to examine how platform providers should set fees for service intermediary platforms and which types of markets they should focus on, not only to acquire a large number of platform users but also to achieve sustainability of the platform ecosystems. We focused on hotel-booking platforms and built an agent-based simulation system of a two-sided market. We constructed our hypotheses based on theories of two-sided markets and e-commerce, and we tested these hypotheses through simulation experiments. The result of the Hypothesis 1 test reveals that the platform fee settings, depending on the balance of supply and demand, could be appropriate in the market of mandatorily purchasing consumers but not for optionally purchasing consumers. Additionally, we showed such platform fee settings could facilitate the acquisition of a large number of platform users; however, these could not improve their sustainability. Moreover, the result of the Hypothesis 2 test indicates that focusing on markets of optionally purchasing consumers with platform fee setting of consumer incentive could accomplish both acquisition of a large number of platform users and sustainability. In these conditions, when the market balance is excessive supply, the degree of market growth showed the highest value within our simulation settings. Conversely, we showed that the improvement in sustainability of the platform-based market was unachievable if the platform focuses on the market of mandatorily purchasing consumers.

In our simulation, we achieved an improvement in sustainability through the increase in profits of both the service providers and consumers, which was caused by the extent of the market size of optionally purchasing consumers when the increase in profits is higher than the decrease in profits caused by the platform fee. Our simulation showed that such an achievement did not occur in the mandatorily purchasing markets. Additionally, the results also show that service providers (or consumers) continued using the platform even when losing profits. Since one side—which is charged a platform fee—is larger than the other, the decision of exiting the platform must include acceptance of losing their transactions. Therefore, we considered a mechanism wherein the participants in the larger side remained at the platform with profit loss in the service intermediary platform-based markets if the platform focuses on the mandatorily purchasing market. Therefore, we imply that this mechanism could lead to failure of the ecosystem due to a collapse of charged participants, even if the platform has acquired a large share of the market.

### 4.1. Theoretical Implications

This study contributes to the research stream of platform fee settings in platform-based markets. Previous studies on pricing of platforms argued that groups with stronger indirect network effects should be charged higher platform fees [20]. Additionally, Rochet and Tirole [15] indicated that platform pricing should be set by the elasticity of demand based on the Ramsey rule. This study referred to such platform fee setting methods and tested whether they acquire a large number of platform users and the sustainability of the platform-based market. Our findings reveal that although it is possible to acquire a large number of platform users, this huge number could not help achieve sustainability. This study suggested that focusing on markets of optionally purchasing consumers is effective in improving the sustainability of the market with consumer incentives rather than fee settings based on the structure of service providers and consumers.

This study also contributes to the research stream on service platform ecosystems in terms of sustainability. Some previous studies have focused on platforms in the service industry. Clemons, Hann, and Hitt [42] focused on airline booking platforms and analyzed price diversification and product differentiation. Zha, Zhang, Yue, and Hua [43] focused on hotel-booking platforms and researched the allocation problem of hotel rooms on platforms. Kung and Zhong [44] focused on the delivery service platform and considered the pricing method of three pricing strategies, namely, membership-based pricing, transaction-based pricing, and cross-subsidization. However, previous studies have not focused on the influence of service platform management on the sustainability of the service industry. We suppose this study is the first step in exploring the sustainability of service platform ecosystems.

### 4.2. Managerial Implications

This study suggests the following implications. As Figure 4 shows, the current platform fee settings of hotel-booking platforms in Japan are mainly combinations of incentive for consumers and charging service providers. Our simulation suggested that the current platform fee setting is appropriate in the case of excessive supply and for a market of optionally purchasing consumers. As Figure 3 shows, since the balance of supply and demand in the Japanese hotel market is mainly excessive supply, our results indicate that the current fee setting should be appropriate to achieve both acquisition of a large number of platform users and market sustainability if the platform focuses on optionally purchasing consumer markets.

Meanwhile, as Figure 2 shows, although about 82% of hotel providers experienced market growth through new customer acquisition from the platform, about 63% considered that platform fees were too high, and about 47% were frustrated by the decline in their profits due to excessive platform fees. Therefore, while we suppose that service platforms can capture some optionally purchasing consumers, the current market size of such consumers cannot satisfy the service providers. Accordingly, we deem the sustainability of the hotel-booking platform ecosystem insufficient, and suggest that platform providers should consider policies and strategies to capture more optionally purchasing consumers.

### 4.3. Limitations and Future Research

This study has several limitations. First, we did not consider aspects of service innovation. One important factor of the platform ecosystem is the emergence of innovation in complementary goods. Previous studies suggested that the introduction of platforms could support the development of goods and the generation of innovation within the ecosystem [1]. Therefore, the service intermediation platforms can not only intermediate existing services but also encourage service providers to focus on service innovation. In doing so, platform providers should capture new customers that need innovative new services to sustain the challenging complementors [10]. Thus, future research could investigate the service innovation aspect of service platform ecosystems.

Second, although this study utilized some data to construct the simulation, we consider that additional data utilization could result in more progressive simulation analysis. For example, we deem that the utilization of dataset from questionnaire surveys from consumers and service providers may be available to investigate the optimal platform fee settings and market situations to improve sustainability. Although the surveys presented in this study investigated the current situation of the Japanese hotel market, future surveys may investigate decision-making mechanisms of participants on the platform to improve the simulation. Additionally, the consumer service selection model was formulated simply as considering the influence of satisfaction of price requirement and that of quality requirement to be equivalent. The utilization of consumer surveys can make such models more realistic. Thus, future research could improve simulations based on questionnaire datasets.

Third, this study supposes two situations of platform use and direct transaction in the simulation. However, decision-making behavior may change between these two situations. For example, when consumers use a platform, the platform's point incentives might lock them in the ecosystem. Conversely,

since the consumers need to reduce search costs at direct transaction situations, certain hotels may capture consumers as loyal customers. Thus, future research could focus on the differences in consumer behavior between platform use and direct transaction.

Fourth, this study focused on the "profitability" of participants as the indicator of sustainability of platform ecosystems. As another significant indicator of sustainability, future studies may consider "long-term stability". In this study, we consider the satisfaction of high profitability indicates long-term stability, since we set the market situation as stable and there were no competitive platforms. However, if we consider more realistic market situations, long-term stability becomes more important. For example, to achieve long-term stability, platform providers may encourage service providers to develop exclusive services on the platform or may actively improve functions and interfaces of their own platforms. Thus, future research could investigate viewpoint of long-term stability as sustainability of the platform ecosystems.

Finally, the analysis and simulation objects in this study were limited to platforms of intermediate hotel services provided by accommodation firms. However, even if it is limited to the lodging market, there are different platforms such as metasearch (e.g., Trivago) and vacation rental intermediation (e.g., Airbnb). Previous studies have not explored how the introduction of these new ventures influences the existing hotel-booking platform ecosystems. Additionally, we believe no studies have yet investigated platform-based market growth and the sustainability of these new platforms. If we consider including these platforms in our research structure, the following may happen. First, Trivago facilitates consumers in selecting the cheapest service across some hotel-booking platforms. This may prompt platform providers to increase incentive programs to acquire consumers on own platforms and, in turn, service providers may suffer by losing more profits. Therefore, the significance of acquiring optionally purchasing consumers may become greater than demonstrated in this study. Second, Airbnb add to the available hotel service providers. Therefore, existing hotel service providers may lose profits with increased competition. However, the service contents and concepts of Airbnb hotel services are somewhat different from existing hotel services. Accordingly, they might co-exist by capturing new customers that can obtain large profits using Airbnb hotel services and not through existing hotel services. Thus, in future, researchers could analyze such new types of platforms in the service industry in terms of their influence, ecosystem growth, and sustainability.

**Author Contributions:** Conceptualization, Y.I., T.T. and K.K.; methodology, Y.I., T.T., K.K.; software, Y.I. and K.K.; validation, Y.I. and K.K.; formal analysis, Y.I.; investigation, Y.I. and T.T.; resources, Y.I.; data curation, Y.I. and T.T.; writing—original draft preparation, Y.I., T.T. and K.K.; writing—review and editing, Y.I., T.T. and K.K.; visualization, Y.I.; supervision, T.T. and K.K.; project administration, Y.I.; funding acquisition, Y.I.

**Funding:** This work was funded by JSPS KAKENHI Grant Numbers 18K12874.

**Conflicts of Interest:** The authors declare no conflict of interest.

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
