# Peer review of "Sustainability of Service Intermediary Platform Ecosystems: Analysis and Simulation of Japanese Hotel Booking Platform-Based Markets"

_sustainability, doi:10.3390/su11174563_

Round 1
Reviewer 1 Report
This paper deals with a platform ecosystem in the service industries, in particular, an online booking platform. The paper focuses on setting the inadequate platform fee settings might lose sustainability of the platform ecosystems. This study has a clear goal as well as a well-developed model and provides fruitful outcomes.
1. The model excludes non-standard hotel booking platforms like Trivago and Airbnb, but many travelers actually depend on those sites (sometimes more than the standard booking platforms). Thus, it may not be fair to exclude these sites. The authors need to provide some ideas about what could happen in their results when they include these sites.
2. There seems to be some misunderstanding about the multi-homing case. Armstrong and Rochet & Tirole do not suggest that participants on the multi-homing side should be charged higher than those in the single-homing side. On the contrary, they argue that multi-homing users should get more favors because they are less likely to be locked in. Please review this point (and the claim (C), too) again.
3. The setting of the simulation model:
3-1 The authors need to explain more about Figure 5. This setting implies a sort of platform dis-intermediation and could be a key feature that affects the simulation outcome. For example, the possible dis-intermediation should be linked with consumers’ search behavior or decision changes. However, I could not see those linkages in the model. Without such linkage, it is necessary to provide (at least) a guess as to what results would be possible if such a linkage were to exist.
3-2 The quality may not be randomly assigned. The service quality is usually closely linked with the price. Thus, it is necessary to provide (at least) a guess as to what results would be possible if both were interrelated.
3-3 Equation (3) needs more explanation. The equation means that user’s payoff or utility is the sum of the price factor and the quality factor. In typical economic modelling, user’s ‘net’ utility is determined by the difference between his/her willingness-to-pay and actual market price (in this sense, this question is also linked with 3-2 above). It may be hard to find other literature (except ones by the same authors) that models the utility like this.
4. Sustainability measure
I’m not sure whether it would be okay with the sustainability measure. The notion of sustainability somehow indicates a long-term issue as well as the healthiness of the ecosystem. Thus, this measure should also reflect both a long-term feature and healthiness of the entire ecosystem. However, the increase and decrease in profits of participants fall short of showing both features. I’d like to see a more appropriate measure for sustainability if presenting a new measure with the current outcomes (data) is not that difficult.
5. The introduction of this paper emphasizes the nature of the platform ecosystem as an open system. Particularly, the paper addresses the importance of complementary participants of the platform ecosystem. However, it is not easy to find an implication of this point from the model and simulation results. Thus, it would be much better to present some ideas interpreting the results from this perspective.
Minor comments
* Some expressions are not natural in English. To name a few, “Moreover, we regard point system for consumers as discount.” in lines 130 and 131; “For the pricing balance between single homing and multi-homing, Rochet and Tirole [14] indicate similar understanding.” in lines 167 and 168; etc.
So, I recommend that a native speaker should proofread the manuscript.
* Typos like (2] in line 290
Author Response
We express our appreciation for your insightful comments, which have helped us significantly improve the paper. We have considered all your comments and revised our manuscript as attached file.

Reviewer 2 Report
The paper that is presented to "Sustainability" is an interesting work, even so it has some considerations to evaluate:
1) The most difficult to interpret in the paper is the relationship between sustainability and platform ecosystem, after reading it several times, I still do not understand this relationship. If this is the object of the work, an effort should be made to clarify it.
2) It is suggested to reduce the abstract (too extensive).
3) Eliminate point 1.1 and leave it alone as, "1. Introduction", it is better and does not infer in bad interpretations. This amendment requires the restructuring of the article, both in points 1.2, 1.2.1 and 1.2.2.
4) I think the structuring of epigraphs of the article is excessive, there comes a point where we do not know where we are and what we want to nalaizar.
The paper needs a re-approach that clarifies much better the objectives that are pursued and what results want to be achieved.
Author Response
We express our appreciation for your insightful comments, which have helped us significantly improve the paper. We have considered all comments and revised our manuscript, as follows. The red parts in the revised manuscript represent the major changes.
[Comment]
1) The most difficult to interpret in the paper is the relationship between sustainability and platform ecosystem, after reading it several times, I still do not understand this relationship. If this is the object of the work, an effort should be made to clarify it.
[Response]
We totally agree with the comment. To clarify the relationship between sustainability and platform ecosystems, we have significantly revised the Introduction as follows.
Platform-based markets have a huge influence on human life in current society. Researchers have developed the concept of platform ecosystems [1,2]. Platform ecosystems are communities orchestrated by platform providers, outside complementors, and consumers [1,2,3,4]. A “business ecosystem” is an “economic community supported by a foundation of interacting organizations and individuals—the organisms of the business world; the member organisms also include suppliers, lead producers, competitors, and other stakeholders” [5,6]. A platform ecosystem restricts its scope to relevant actors, such as the platform itself and its providers and users. However, platform ecosystems do not generally restrict participation or withdrawal. Therefore, although a platform ecosystem has boundaries, it is an open system; that is, its composition may not mandatorily converge to any specific states because of outside interaction [7].
In platform ecosystems, complementors develop and/or provide complementary goods (products and/or services) using platform technology. Consumers then purchase these complementary goods via the platform. A platform ecosystem induces consumers with various needs to adopt the platform [8]. The success of a platform ecosystem depends upon the success of the entire ecosystem [9].
Actually, some previous studies reported examples of platform ecosystems that declined earlier than did those of competitors, although it succeeded in acquiring a large number of platform users. For example, the Nintendo Wii video game platform ecosystem declined earlier than its competitor (PlayStation 3) due to the profitability of complementors (video game software providers) [10]. Furthermore, Nintendo DS declined faster than did PlayStation Portable because of similar reasons [11]. Thus, in the research on platform ecosystem, we consider both its sustainability and growth as significant factors. Nevertheless, such research is insufficient.
Thus, this study examines the sustainably of platform ecosystems. In particular, we focus on service intermediary platforms. Typical examples of such platform are Expedia and Hotel.com, which function as intermediates between hotel service providers and travelers as consumers. Such platforms can improve matching between consumers and services by allowing consumers to find preferable services with ease. If a platform can acquire sufficient consumer base in the market, the service providers become interested in joining it, with an increasing expectation of obtaining numerous customers. The platform providers profit by charging fees for the use of platform during transactions.
As noted, one reason for the loss of sustainability in platform ecosystems is the profitability of its participants. General participants would not be motivated to remain with the platform at low profits. Therefore, one of the most significant factors for sustainability is profitability for complementors and consumers. At this point, we deem that service platforms may have two factors influencing sustainability, which are may not occur in hardware-software type platforms.
First, the earnings model of platform providers may be different. In case of hardware-software type platforms, consumers have the incentive to purchase hardware, because generally, some parts of the software can be exclusive to the platform. Therefore, the platform provider can obtain profit from consumers. However, at least some current service intermediary platforms are different. For example, hotel booking platforms typically handle existing hotel services. Consumers can use the same hotel services when not using the platforms. Therefore, we consider charging platform fee to consumers is difficult; the platform provider will gain profit from complementors (hotel service providers). We confirm that the fee settings of current hotel booking platforms correspond to those in subsection 2.1.2. Second, service providers have limitations in producing goods in comparison to product providers, because of the inseparability of production and consumption [12]. Thus, charging a platform fee for service providers can have a larger negative impact on their profitability than it does on product providers. In fact, some hotel service providers experienced declining profitability when using hotel booking platforms, as subsection 2.1.1 shows. Thus, the current structure of service intermediary platforms may essentially become unsustainable ecosystems in terms of risk associated with profitability of service providers.
We now focus on the sustainability of service intermediary platform ecosystems. Considering that the platform fee setting is insufficient, we focus on market types, in which the service intermediary platforms can have large value. Thus, to achieve both large platform users and sustainability of the service intermediation platform ecosystems, this study aims to clarify appropriate platform fee settings and promising market types. We investigate hotel booking platforms as representative of service intermediary platforms. This study adopts an agent-based simulation approach to its analysis.
------------------------------------------------------------------------------------------------------------------
[Comment]
2) It is suggested to reduce the abstract (too extensive).
[Response]
We have reduced the abstract by approximately 40 words to mitigate extensiveness.
“Abstract: To achieve both large platform users and sustainability of the service intermediation platform ecosystems, this study attempts to clarify appropriate platform fee settings and promising market types. It focuses on hotel booking platforms and platform-based markets. We conduct agent-based simulation experiments with the conditions of (a) platform fee setting, (b) supply and demand balance, and (c) consumer type categorized as “mandatorily” purchasing consumers (e.g., business travelers, whose purchase of any hotel service is mandatory) and “optionally” purchasing consumers (who can refuse services if their requirements are not satisfied). Our simulation results reveal that when the platforms focus on mandatorily purchasing consumers, they could acquire larger platform users through fee settings, depending on the balance of supply and demand; however, they could not maintain ecosystem sustainability in any case. When the platform focuses on optionally purchasing consumers, it can achieve both large platform users and ecosystem sustainability with platform fee settings of consumer incentives, especially in excessive supply markets.”
------------------------------------------------------------------------------------------------------------------
[Comment]
3) Eliminate point 1.1 and leave it alone as, "1. Introduction", it is better and does not infer in bad interpretations. This amendment requires the restructuring of the article, both in points 1.2, 1.2.1 and 1.2.2.
[Response]
We have eliminated point 1.1 and written it as “1. Introduction.” We also moved the contents of point 1.2 and 1.3 to section 2. Since section 2 is now longer, we have added explanations for its structure as follows.
“The structure of this section is as follows. In the first subsection, to understand current situation of hotel booking platform ecosystems, we present the results of investigation of such platforms in Japanese hotel markets. In the second subsection, we explain the hypotheses of this study. In the third subsection, we present an overview of our agent-based simulation system. We explain the detailed settings of agents in the fourth subsection. The final subsection describes the simulation experimental settings.”
------------------------------------------------------------------------------------------------------------------
[Comment]
4) I think the structuring of epigraphs of the article is excessive, there comes a point where we do not know where we are and what we want to nalaizar.
[Response]
We have now excluded the following epigraphs and revised the related text.
1.3.1. Platform fee setting depending on the balance of supply and demand
1.3.2. Focus on markets for optionally purchasing consumers
2.1.1. Basic settings
Since excluding further epigraphs will make the text too long in certain sections, we have not done so.